# A Method Based on Curvature and Hierarchical Strategy for Dynamic Point Cloud Compression in Augmented and Virtual Reality System [note 1]

**DOI:** 10.3390/s22031262

**Published:** 2022-02-07

**Authors:** Siyang Yu, Si Sun, Wei Yan, Guangshuai Liu, Xurui Li

**Affiliations:** 1School of Electronic, Electrical and Communication Engineering, University of Chinese Academy of Sciences, Beijing 100049, China; yusiyang@ioe.ac.cn; 2Institute of Optics and Electronics, Chinese Academy of Sciences, Chengdu 610209, China; sunsi@ioe.ac.cn; 3School of Mechanical Engineering, Southwest Jiaotong University, Chengdu 610031, China; motorliu7810@swjtu.edu.cn (G.L.); xuruili@my.swjtu.edu.cn (X.L.)

**Keywords:** dynamic point cloud, curvature-based, hierarchical refine, video-based compression

## Abstract

As a kind of information-intensive 3D representation, point cloud rapidly develops in immersive applications, which has also sparked new attention in point cloud compression. The most popular dynamic methods ignore the characteristics of point clouds and use an exhaustive neighborhood search, which seriously impacts the encoder’s runtime. Therefore, we propose an improved compression means for dynamic point cloud based on curvature estimation and hierarchical strategy to meet the demands in real-world scenarios. This method includes initial segmentation derived from the similarity between normals, curvature-based hierarchical refining process for iterating, and image generation and video compression technology based on de-redundancy without performance loss. The curvature-based hierarchical refining module divides the voxel point cloud into high-curvature points and low-curvature points and optimizes the initial clusters hierarchically. The experimental results show that our method achieved improved compression performance and faster runtime than traditional video-based dynamic point cloud compression.

## 1. Introduction

Since the twenty-first century, the development of three-dimensional (3D) sensing technology has set off a wave of innovation in Augmented and Virtual Reality Production (AR/VR). In 2007, Google introduced the Street View, which allowed users to view and navigate the virtual world, and now there is a complete version that is commercially available [1]. In 2014, Samsung released the Samsung Gear VR for Galaxy smartphones to gain a completely untethered, easy-to-use experience [2]. In 2020, Coronavirus Disease catalyzed AR retail and VR conferences. Meanwhile, the use of 3D point clouds to represent real-world scenarios in an immersible fashion for AR/VR system has become increasingly popular in recent decades. Apple brought point cloud to mobile devices in 2020 and successfully created a more realistic augmented reality experience [3]. In 2021, Zhongxing Telecom Equipment built the AR Point Cloud Digital Twin Platform and said that the platform had formed a scaled deployment and demonstration effect [4]. Moreover, point clouds have achieved significant success in many areas, e.g., visual communication [5], auto-navigation [6], and immersive systems [7]. Usually, a point cloud comprises its geometric coordinates and various attributes such as color, normal, temperature, and depth. It always comes with a large amount of data, which leads to the heavy transmission overload of today’s networks. Unlike the 3D mesh models, point clouds do not contain any edges and facets. More challenging tasks (high-density, high-complexity, and high-resource requirements) have increased downstream processing. In short, before employing the point cloud to AR/VR, it is important to compress it first.

Recently, various families of 3D Point Cloud Compression (PCC) methods have been proposed [8,9,10,11,12,13]. The first step is to set up a regular structure for point clouds to deal with the massive discrete points. Garcia et al. [12] presented a lossless static PCC project based on octree structure. In addition, the value and position of the parent node was used to generate a set of contexts for entropy coding. Besides, clustering is also a superior technology used in PCC because it can take full advantage of point cloud characteristics. Sun et al. [14] divided a lidar point cloud into the ground and main objects. They saved the shape of each divided region as a contour image and the corresponding value as a predicted value. Yuan et al. [15] discarded the unimportant points to reduce the amount of data passed into the encoder and maintained the point cloud accuracy through upsampling.

The above methods only consider spatial redundancy and ignore the temporal domain. In actual situations, the change usually occurs in local areas for dynamic context, so the temporal correlation within the adjacent point clouds can be expected [16,17,18]. Kammerl et al. [17] took advantage of a differential octree representation that saves computation and reduces memory requirements. Moreover, Garcia et al. [18] applied an incomplete low-resolution version of the current frame and a full-resolution version of the previous frame to compress dynamic point clouds.

The Joint Photographic Experts Group (JPEG) divided the scene into foregrounds and background, then used different coding formats to make them coexist in the same setting [19]. However, the corresponding standard reference software, which was the integration of the algorithms of each module, did not consider the calculation complexity, so it was not suitable for application scenarios with high real-time requirements. Moreover, the standardization of point cloud data compression, holographic data compression, and quality test procedures is still in the exploratory stage [20]. It is noteworthy that, in 2018, the Moving Picture Experts Group (MPEG) announced a standard for PCC [21] which divided point cloud models into three categories: static; dynamic, time varying point clouds used for immersive video and AR video; and dynamically acquired, used in mobile mapping. Later, the Geometry-Based PCC method (G-PCC) for static and dynamically acquired situations and the Video-Based PCC method (V-PCC) for dynamic conditions were grown. The V-PCC, which makes the best of the mature High Efficiency Video Coding (HEVC) 2D video coder practically, is favored by immersive applications.

Due to the functionalities and optimizations accumulated in video codecs, there are many improvements in the literature for video-based PCC [22,23,24,25,26,27,28,29]. Sevom et al. [24] reduced the encoding and decoding complexity by dropping secondary texture and geometry content from the processing chain and introducing the geometry-guided 3D data interpolation. Li et al. [25] used reconstructed geometric information to improve the coding efficiency of attribute videos in the dynamic PCC. In the field of geometric compression, Xu et al. [26] conducted the geometry PCC via patch-wise polynomial fitting. However, this method was not satisfactory at high bit rates. Costa et al. [27] proposed several patch packing methods, including packing algorithms and associated sorting and positioning metrics, without compromising in any way the V-PCC stream. Moreover, Lopes et al. [28] used projection planes adapted to point cloud characteristics for patch generation. However, this method did not use the correlation between adjacent frames. Instead, Junsik et al. [29] estimated the motion vector of the dynamic condition using temporal correlation, yielding greater benefits than conventional video encoders, but its focus is not on the patch generation phase. 

Most improvements still use a limited refining process with a fixed search radius independent of point cloud characteristics and carry out neighborhood searches for all points, seriously impacting the encoder’s runtime. To tackle this, we propose an improved PCC method based on curvature estimation and hierarchical operation. The core idea is to use curvature-based hierarchical refinement and segmentation technology before the point cloud enters the 2D space to reduce the algorithm’s complexity while continuing the superior performance of the video encoder. The main contributions of this manuscript are as follows:An innovative curvature-based hierarchical dynamic PCC method is proposed to reduce the time complexity of the refinement operation and even the overall framework;The decision-making methods of voxel size and the number of iterations are proposed to obtain a good trade-off between compression quality and compression time.

The remainder of this paper is organized as follows.

Following this introduction, we describe the curvature-based hierarchical dynamic PCC method in detail in Section 2. In Section 3, we selected Redandblack, Soldier, Longdress, Loot, Basketball_player, Dancer, Ricardo and Phil eight sequences, presenting the comparison results between the proposed techniques and the classic video-based algorithm. The experiments showed that the proposed scheme can reduce the overall runtime by 33.63% on average, with clear Rate-Distortion (R.D.) performance benefit and subjective effect improvement. Finally, we conclude the entire paper in Section 4.

## 2. The Curvature-Based Hierarchical Dynamic Point Cloud Compression Method

In the past decades, the video-based theory has been proven to be a low-cost solution for promoting point cloud applications. This paper first utilizes normal similarity between real point and pre-orientated elemental planes to segment the point cloud frame. Next, curvature characteristic is used to hierarchically optimize the clusters of the first step. Then, these refined clusters are packaged into images, such as a geometric image and a color image. Finally, all images can be compressed with the existing versatile video codec. On account of the realization that the latter two steps can refer to the current literature, the proposed method can be summarized as the following three main steps: initial segmentation, curvature-based hierarchical refinement, and post-processing (including image generation and video-based coding).

### 2.1. Initial Segmentation

The premise of using 2D video coders is that the point cloud model is mapped to a 2D plane in a simple but efficient way. Projecting the model onto the six planes of its bounding box is given priority.

More precisely, we first calculate the normal vectors of all points in the point cloud. Then, the point cloud is divided into six basic categories using the six planes of the pre-defined unit cube, which can be represented by normal vectors as: (1, 0, 0), (0, 1, 0), (0, 0, 1), (−1, 0, 0), (0, −1, 0), (0, 0, −1), as shown in Figure 1. The segmentation basis is the similarity of the normal vectors between the real point and the orientation plane, maximizing the dot product of the point’s normal vector and the plane’s normal vector. The corresponding pseudo-code is Algorithm 1:
**Algorithm 1** The pseudo-code of initial segmentation
**Input:** the normal of each point N,  the normals of orientation planes Plane,  the number of orientation planes *numPlanes*,  the number of points *numPoints*
**Output:** initial clusters partition  **For**
i=0 to *numPoints*-1 **do**
  bestScore = 0    **For** j=0 to *numPlanes*-1 **do**      scoreNormal = N[i]·Plane[j]      **If** scoreNormal > bestScore **do**        bestScore = scoreNormal        clusterIndex = j
      **End If**
    **End For**    partition[i]=clusterIndex
  **End For**

Segmentation aims to find some patches that satisfy time-coherent, low-distortion, and are convenient for dimensionality reduction. Maximizing time-coherent and minimizing distance and angle distortion is beneficial to the video encoder to fully use the spatio-temporal correlation of point cloud geometry and attribute information. However, the previous work does not guarantee a good reconstruction effect due to auto-occlusions. To avoid it, we then refine the clustering results.

### 2.2. Curvature-Based Hierarchical Refinement 

The input of the refine module is the clustered geometric coordinates and attributes. And we look forward to getting some clusters meeting the image or video compression technology requirements, such as smooth borders and less occlusion. Patches with soft edges are incredibly effective in succeeding geometric filling and attributing filling parts, which inspired us to consider adjacent points in the later refining process. Less occlusion means avoiding the projection of different points onto the same location as far as possible by controlling the projection direction. More collisions will cause more information loss, so the normal vector of the projection plane is also deemed as a considerable factor.

Figure 2 provides an overview of the curvature-based hierarchical refinement, which is a friendlier refining solution to reduce computational complexity and runtime and will not alter the bitstream syntax and semantics or the decoder behavior.

The main steps cover partitions of geometric coordinate space, neighborhood information search, and clustering based on scores. It should be noted that the neighborhood information search comprises the calculation of curvature and the hierarchical search; the final scores extrapolate from the normal vector score, voxel-smooth score, and smooth score.

#### 2.2.1. Partitions of Geometric Coordinate Space

Even if only ten neighbors are searched for each point, the clusters updating for whole cloud containing hundreds of thousands of points is laborious. As a result, the refining procedure was suggested to be simplified by adding a voxel-based constraint to the neighborhood search [22]. Inspired by this, we first use uniform voxels to divide geometric space, then perform a neighborhood search on the voxelized point cloud instead of the entire model. The first traversed point in each voxel is selected to identify the voxel instead of the geometric center point due to the following calculations being for integers. Specifically, the coordinates of identifying individual and contained points for each voxel are stored. In the searched nearest-neighboring filled voxels, all interior points that meet the distance limit are regarded as the neighborhood information of the current voxel.

The more extensive the voxel size, the fewer points need to be considered in each neighborhood search and the fewer identification points are explored, which means the difference between the searched neighbors and the actual situation is more significant, as shown in Figure 3.

#### 2.2.2. Neighborhood Search for Each Filled Voxel

The above partition ideal provides convenient conditions for neighborhood search. However, redundant calculations may occur if all regions are considered equivalent and use the fixed search radius, as shown in Figure 4. If different clusters are marked with diverse colors, it can be found that there are plenty of points belonging to the same group in the chest and abdomen of the Loot point cloud. Wherein the clustering index is updated iteratively, the renewal results obtained by various search radii are consistent because all neighbors have identical indexes. Accordingly, it is reasonable to use a smaller search radius for some local regions.

Considering that the update of the clustering index is related to the neighbors and inseparable from the normal vectors, the curvatures originating from normal vectors on local surfaces are selected to study the search radius. The points of low curvature located on a virtually flat surface need bits of neighbors to reflect the index-change trend. By comparison, the points with high curvature, which have an apparent diversification in the normals between them and adjacents, call for a more comprehensive neighborhood search to prevent the normals from unduly influencing the final result, as shown in Figure 5. In the intra-cluster, the small-scale neighborhood search increases the number of patches for high-curvature regions but does not affect low-curvature parts. On the contrary, the small-scale neighborhood search results in patches with sharp edges for high-curvature areas at cluster boundary but has little impact on low-curvature areas. Therefore, the search for nearest-neighboring-filled voxels in this paper was completed based on curvature grading to reduce the amount of calculation. Low-curvature zones implement a small-scale neighborhood search, while high-curvature zones conduct a large-scale neighborhood search.

We apply the local surface fitting method to calculate curvatures. Firstly, principal component analysis is used to estimate normal vectors. Then, a minimum spanning tree is used to orient the normal vectors. The normal vector of $P_{i}$ is used as the vertical axis to establish a new local coordinate system. Finally, a quadric surface fitting is performed on the local coordinate system, and its fitted surface parameters are used to estimate the curvature at $P_{i}$, as
(1)$$K=(K_{1}+K_{2})/2,$$
where $K_{1}$ and $K_{2}$ are the two principal curvatures of $P_{i}$, respectively.

The predictions of curvatures are based on the identification points instead of the entire cloud model, as the number of voxels is far less than the number of actual points. In Figure 1, the curvatures histogram of the identification points shows that only a few individuals had a relatively high curvature as most areas had excellent flatness. Consequently, all identification points are classified into low-curvature points and high-curvature points, according to the low-curvature determination ratio, as shown in Figure 6. Then, we can estimate the neighborhood information hierarchically to reduce complexity.

#### 2.2.3. Clustering Based on Final Score

For points with different curvature grades, various search radii are used. Traverse all the identification points and the nearest-neighboring voxels for each identification point are stored in a vector and used to compose the final score.

The normal vector score, which refers to the influence of the normal vectors on the clustering index, must be considered first to get a patch with less occlusion. These six projection planes, as mentioned above, are also used to estimate the normal vector score in the refine segmentation process, according to
(2)ScoreNormal[i][p]=normal[i]·orientation[p],
where normal[i] is the normal vector of the *i*-th point, and orientation[p] is the normal vector of the *p*-th projection plane.

The projection results for different planes are shown in Figure 7. The greater the normal vector score, the fewer points projected to the same position and the fewer collisions occur. Therefore, the calculation of normal vector scores is necessary for each iteration.

The neighbors of each point will also affect its corresponding clustering index to avoid uneven boundaries, i.e., the smooth scores, which refers to the influence of adjacent points on the final clustering index. If the number of neighbors corresponding to $P_{i}$ is $n{umNeighbors}_{i}$, the amount of computation for smooth scores is
(3)$$AmountComputation=\sum_{i=0}^{N-1}n{umNeighbors}_{i},$$
where *N* is the number of identification points.

The appraisal of the smooth scores is undoubtedly demanding but can be simplified by the accumulation of the voxel-smooth score thanks to the partition of geometric coordinate space. The voxel-smooth score for each filled voxel associated with each projection plane needs to be computed first by counting the number of points in the voxel, which are clustered to the projection plane during the refining process. Then, the smooth score can be defined as
(4)v=FilledVoxels[i],
(5)$$ScoreSmooth[v][p]=\sum_{j=1}^{size(nnFilledVoxels)}voxScoreSmooth[nnFilledVoxels[v_{j}]][p],$$
where v is the index of the voxel containing the *i*-th point and p is the projection plane index; $nnFilledVoxels[v_{j}]$ is the *j*-th neighboring voxel and voxScoreSmooth is the set of voxel-smooth scores for all the adjacent voxels. Hence, the smooth score would be identical for all points inside a voxel.

The normal vector score and the smooth score, as mentioned above, are combined to determine the final clustering index through a weighted linear combination, as
(6)$$score[i][p]=ScoreNormal[i][p]+\frac{\lambda }{size(nnFilledVoxels[v])}ScoreSmooth[i][p],$$
where λ is the influence coefficient of the smooth score on the final score, and its value is specified by [30]. After clustering each point to the projection plane having the highest final score as calculated at Equation (6), the cluster update is completed once. 

In the refining process, the total number of loops would be
(7)$$\begin{array}{ll} totalLoop= & numPoints\times numPlanes+\\ & maxNumIters\times \sum_{i=0}^{N-1}\left(numPoints+numPoints\times numPlanes\times numNeighbors_{i}\right), \end{array}$$
where maxNumIters is the maximum number of iterations and numPlanes is the number of projection planes in the refinement. A small-scale neighborhood search is used for most points in the proposed method, so the $numNeighbors_{i}$ for most points is lower than that of the video-based method. Therefore, the total number of loops diminishes, which successfully reduces the computational complexity.

In summary, the pseudo-code for refinement is Algorithm 2:
**Algorithm 2** The pseudo-code of refinement**Input:** initial clusters *partition*  the normal of each point N,  the normals of orientation planes Plane,  the number of orientation planes *numPlanes*,  the number of points *numPoints*  the maximum number of iterations *maxNumIters*  the impact factor *λ*    the number of neighbor voxels used for the search *nnFilledVoxels*  voxel list *V***Output:** updated clusters partition  ω=λ/numNeighbors  **For**
i=0 to *numPoints*-1 **do**    **For** j=0 to *numPlanes*-1 **do**      scoreNormal[i][j] = N[i]·Plane[j]    **End For**  **End For**  **For**
k=0 to *maxNumIters* -1 **do**    **For**
i=0 to V.size-1 **do**      voxScoreSmooth[$V_{i}$][ ] = 0      **For** j=0 to $V_{i}.size$-1 **do**        voxScoreSmooth[$V_{i}$][$partition[V_{i}[j]]$]++//calculate the voxel smooth score      **End For**    **End For**    **For**
i=0 to *numPoints*-1 **do**      **For** j=0 to *numPlanes*-1 **do**        v=filledVoxels[i]//the voxel at which the *i*-point is located        Find the neighboring voxels and store them in *nnFilledVoxels*        scoreSmooth[i][j] = 0        **For** n=0 to *nnFilledVoxels*.size-1 **do**          scoreSmooth[i][j] += voxScoreSmooth[nnFilledVoxels[n]][j]        **End For**      score[j]= scoreNormal[i][j] + ω•scoreSmooth[i][j]      **End For**    Cluster the point to j which related score is the highest    tmpPartition[i]=clusterIndex    **End For**    partition[i]=clusterIndex  **End For**

### 2.3. Post-Processing: Image Generation and Video-Based Coding

Consistent with classical video-based methods, the connected component extraction algorithm is applied to extract the patches obtained by curvature-based hierarchical refinement, and then we map the connected components to a 2D grid. The mapping process needs to minimize the 2D unused area, and each grid belongs to only one patch. The geometric information of the point cloud is stored in the grid to generate the corresponding 2D geometric image. Similarly, the attribute image can also be easily obtained. To better handle the case of multiple points being projected to the same pixel, the connected component can be projected onto more than one image. 

Finally, the generated images are stored as video frames and compressed using the video codec according to the configurations provided as parameters. Details are available in [21,30].

## 3. Performance Assessments

### 3.1. Test Materials and Conditions

We carried out many tests using dynamic point clouds captured by RGBD cameras. The Redandblack, Soldier, Longdress, and Loot four sequences in MPEG 8i Dataset [31] are complete point clouds with a voxel size close to 1.75 mm and a resolution of 1024 × 1024 for texture maps; the basketball_player and dancer two sequences in Owlii Dataset [32] are complete point clouds with a resolution of 2048 × 2048 for texture maps; the Ricardo and Phil are two other sequences in Microsoft Voxelized Upper Bodie’s Dataset [33] and are frontal upper body point clouds with a voxel size compact to 0.75 mm and a resolution of 1024 × 1024 for texture maps, as shown in Figure 8. The point cloud is a set of points (*x*, *y*, *z*) constrained to lie on a regular 3D grid. In other words, it can be assumed to be an integer lattice. The geometric coordinates may be interpreted as the address of a volumetric element or voxel. The attributes of a voxel are the red, green, and blue components of the surface color. Note that each series takes 32 frames for experimentation and comparison.

Select the most popular V-PCC as the comparison item to analyze the advantages and disadvantages of the proposed method. For better evaluation of the R.D. quality, the Bjontegaard Delta-Rate (BD-rate) and Bjontegaard Delta Peak Signal-to-Noise Ratio (BD-PSNR) metrics [34] are calculated, which makes the comparison of different compression solutions possible when considering several rate-distortion points. The PSNR, which aims to report the distortion values, is calculated as (8)$$\begin{array}{cc} PSNR_{color}=10\log_{10}(\frac{{\left(p_{color}\right)}^{2}}{MSE}), & PSNR_{geometry}=10\log_{10}(\frac{3p^{2}}{MSE}) \end{array},$$
where p and $p_{color}$ are the peak constant value for geometric distortions and color distortions of each reference point cloud, respectively, and MSE is the mean squared error. Based on this, BD-rate is defined as the average difference between the area integral of the lower curve divided by integral interval and that of the upper curve separated by the integral interval:(9)BD−rate=1D−HDL∫DLDH(r2−r1)dD,
where $r=a+bD+cD^{2}+dD^{3}$, r=log(R), *R* is the bitrate; *a*, *b*, *c*, and *d* are fitting parameters; *D* is the PSNR; DH and $D_{L}$ are the high and low end, respectively; $r_{2}$ and $r_{1}$ are the curves. A negative BD-rate indicates that the encoding performance of the optimized algorithm has been improved. On the other hand, BD-PSNR expresses the promotion in the objective quality at the same rate, which is given as
(10)BD−PSNR=1r−HrL∫rLrH(D2(r)−D1(r))dr,
where $D=a+br+cr^{2}+dr^{3}$, rH, $r_{L}$, $D_{2}(r)$ and $D_{1}(r)$ are the highest logarithm of bitrate, the lowest one, the original curve, and the compared curve, respectively. The larger the BD-PSNR, the better the proposed algorithm. Furthermore, without losing fairness, experiments strictly implement the common test condition for dynamic PCC provided by MPEG [24].

### 3.2. Voxel Size and the Number of Iterations

Table 1 provides the BD-rate, BD-PSNR, and runtime savings for V-PCC with a voxel size of four compared to a voxel size of two. As the voxel size increases, the encoder’s runtime is reduced by 53.73% on average, but the related geometric and color quality suffers a severe loss. In detail, the D1 bitrate increases by an average of 2.78%, while the Y bitrate rises by an average of 3.32%. MPEG explains that the large voxel size is more suitable for real-time applications because high-precision reconstruction is not required in this case. However, we are inclined to promote the real-time capability of compression schemes while ensuring high quality. Therefore, we focus on weakening complexity that retains a great deal of quality based on the voxel size equal to two.

Figure 9 describes the impact of the maximum number of iterations on the geometric performance, color performance, and runtime. Results display that although the number of iterations increased eight times, D1-PSNR only increased by less than 0.1%. In terms of color, the compression result after ten iterations was not the worst, and the outcome of 90 iterations was not the best. Meanwhile, the cost of time steadily increased. In summary, the rise in the number of iterations has little impact on geometric performance, unstable attribute optimization, and time. Consequently, we directly suggest lessening the value in [24], which is 10.

### 3.3. The Low-Curvature Ratio and Search Radius

According to the common test condition, the search radius for high-curvature points is set to 96. The other radius suitable for low-curvature points needs to be further analyzed, as shown in Figure 10 and Figure 11. We used two types of point clouds for experiments.

The R.D. performance is built up when the radius decreases slightly but declines sharply with further decrease. Even when the radius is less than 25, both the geometry and the color information show a huge loss. Regarding time consumption, when the radius is greater than 36, the saving rate of encoder’s runtime is not more than 30%. Hence, the radii equal to 25 or 36 are substituted into the analysis of low-curvature ratio, as shown in Figure 10. The radius of 25 is outstanding on time but poor on quality. On the contrary, no matter what the low-curvature ratio is, the compression result of the encoder with a radius of 36 is satisfactory. Considering that the ultimate goal is to reduce the runtime, the low-curvature ratio is set as 0.92 and the search radius of low curvature is set as 36.

### 3.4. R.D. Performance Evaluation

All point clouds have a similar change trend on their R.D. curves. Figure 12 provides a detailed performance description of *Redandblack* on geometry and color. The curvature-based hierarchical method has a performance like V-PCC at a low bpp, but less costly and with better quality at a high bpp. It is undeniable that the proposed method is valid for the dynamic condition.

From Table 2 and Table 3, our method is much better than V-PCC, with a voxel size of four in geometry and color performance. However, there is also an average 50.52% increase in runtime due to the reduction of voxel size which enhances the complexity of the neighborhood search and the determination of final score. This is negligible in most applications that do not have extreme real-time constraints. At the same time, data compared with a voxel size of two shows that the proposed approach improves performance and efficiency obviously, saving an average of 33.63% of the total time. After the tradeoff between real-time performance and accuracy, our method can achieve clear quality improvement and, in most cases, shorten the encoder’s runtime.

Besides, the visual effects of our method, as compared with V-PCC, are demonstrated in Figure 13, Figure 14 and Figure 15. Clearly, the point clouds compressed by a voxel size of four are generally unsmooth and have apparent cracks. The point clouds condensed by a voxel size of two outperform voxel size of four, but there are also some cracks. The point clouds conducted by our approach are closest to the original point clouds. Therefore, the method proposed in this paper can achieve better visual effects than V-PCC, consistent with the results obtained from the analysis of R.D. performance, as described earlier.

## 4. Conclusions

In this paper, we proposed an improved dynamic PCC method based on curvature estimation and hierarchical strategy to reduce the runtime of the video-based compression scheme and obtain an apparent quality gain. Firstly, the proposed method segments original data into six primary clusters utilizing normal similarity. Secondly, we suggested a curvature-based hierarchical refining approach to optimize clusters. Finally, image generation technology and video codec were used to map point cloud to 2D image and compression. 

The curvature-based hierarchical method’s specific flow can begin with generating voxelized identification points by the partition of geometric coordinate space. Next, classify identification points into low-curvature points and high-curvature points. Then, estimate the neighboring voxels and final scores hierarchically. Last, converge each point to the cluster associated with the highest score to obtain patches with smoother boundaries and fewer repeated points. 

The experimental results show that the proposed scheme can save 33.63% compression time on average, with clear R.D. performance benefits and subjective effect boost, and is suitable for most AR/VR applications. However, the curvature-based hierarchical method requires only the characteristics of geometric space. In future work, we will consider using the properties of the attribute space to improve compression quality.

## Figures and Tables

**Figure 1 sensors-22-01262-f001:**
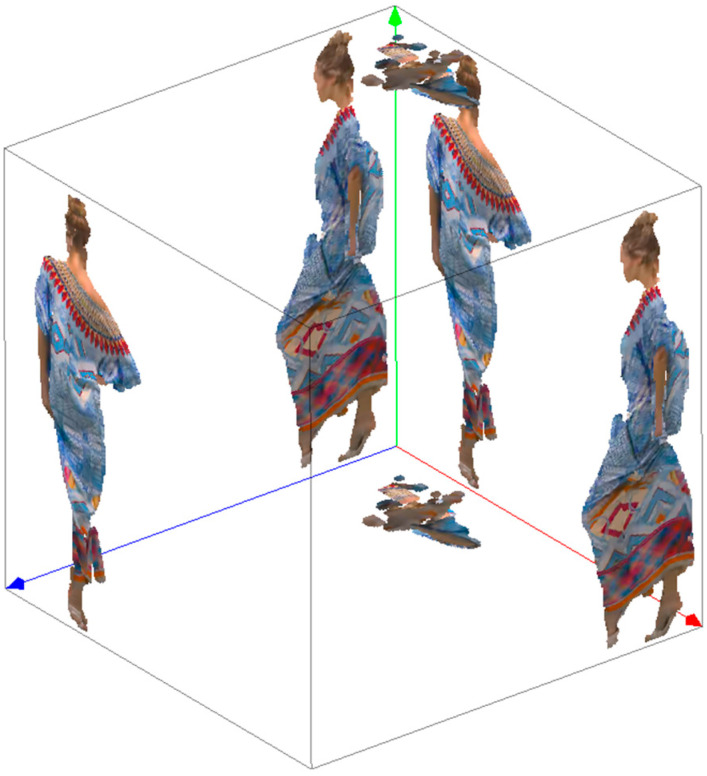
Point cloud projected onto “bounded-box” planes. MPEG. V-PCC codec description. In 127th MPEG Meeting, Gothenburg, Sweden, 8–12 July 2019; *ISO/IEC JTC1/SC29/WG11 (MPEG) output document N18674*.

**Figure 2 sensors-22-01262-f002:**
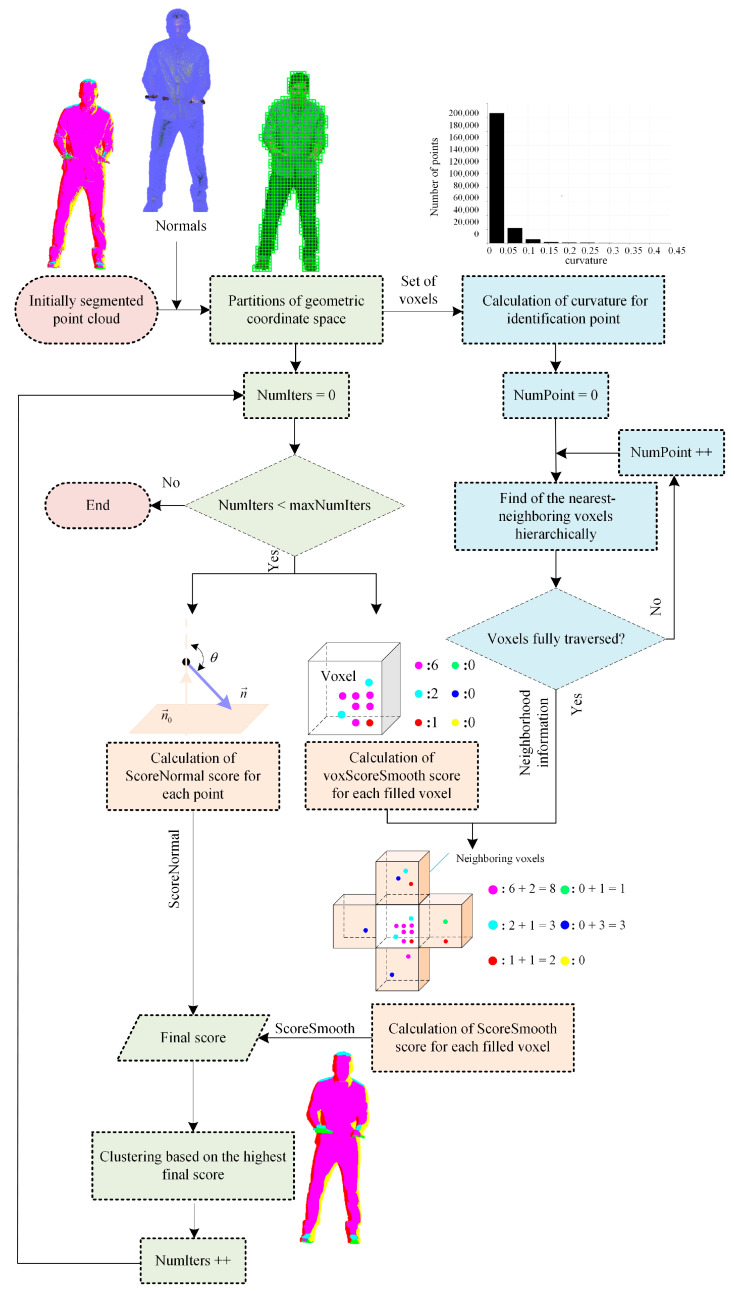
Curvature-based hierarchical refinement architecture.

**Figure 3 sensors-22-01262-f003:**
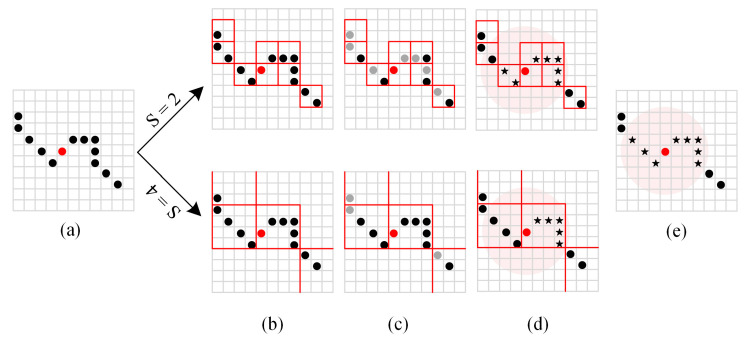
Neighbors searched in different sizes of voxel (S refers to the voxel size): (**a**) search center marked with red; (**b**) voxel division; (**c**) identification points; (**d**) the searched neighbors marked with asterisks; (**e**) actual adjacent points marked with asterisks.

**Figure 4 sensors-22-01262-f004:**
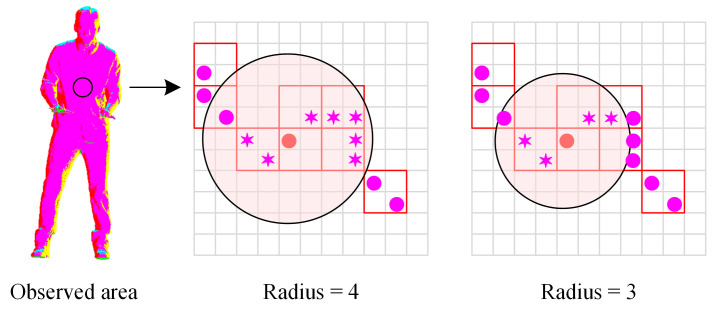
Local neighborhood information obtained by various radii. Note that the searched neighbors are signaled with asterisks.

**Figure 5 sensors-22-01262-f005:**
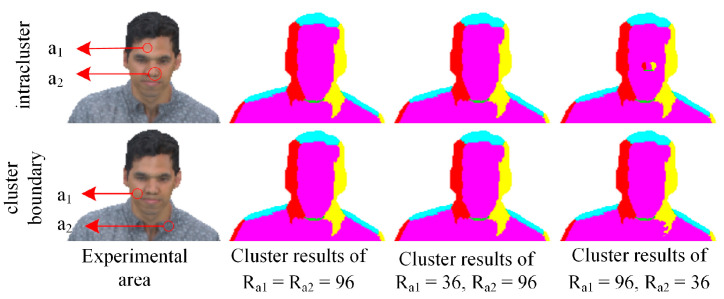
The clustering results obtained by searching with various radii in a specific area (a1 is a low-curvature area, while a2 is a high-curvature area).

**Figure 6 sensors-22-01262-f006:**
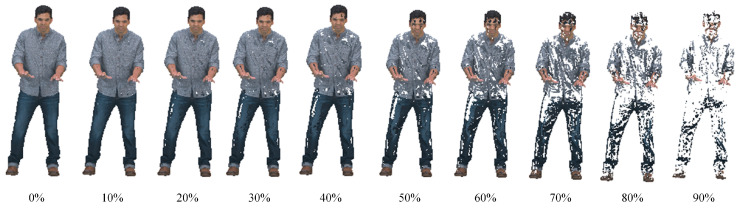
High-curvature points under various low-curvature determination ratios.

**Figure 7 sensors-22-01262-f007:**
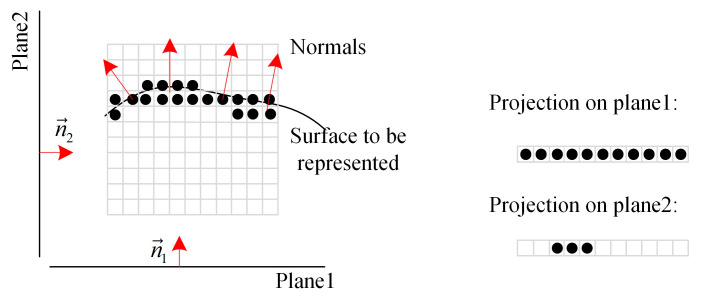
Projection to different planes.

**Figure 8 sensors-22-01262-f008:**
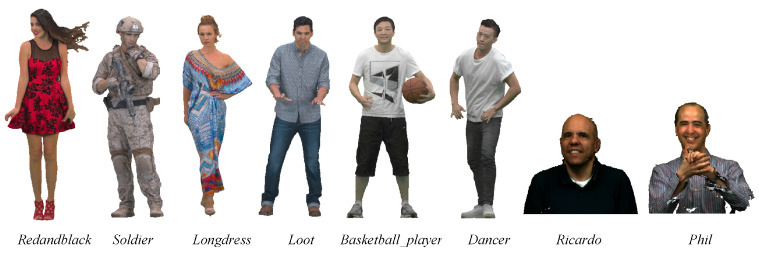
The original point clouds.

**Figure 9 sensors-22-01262-f009:**
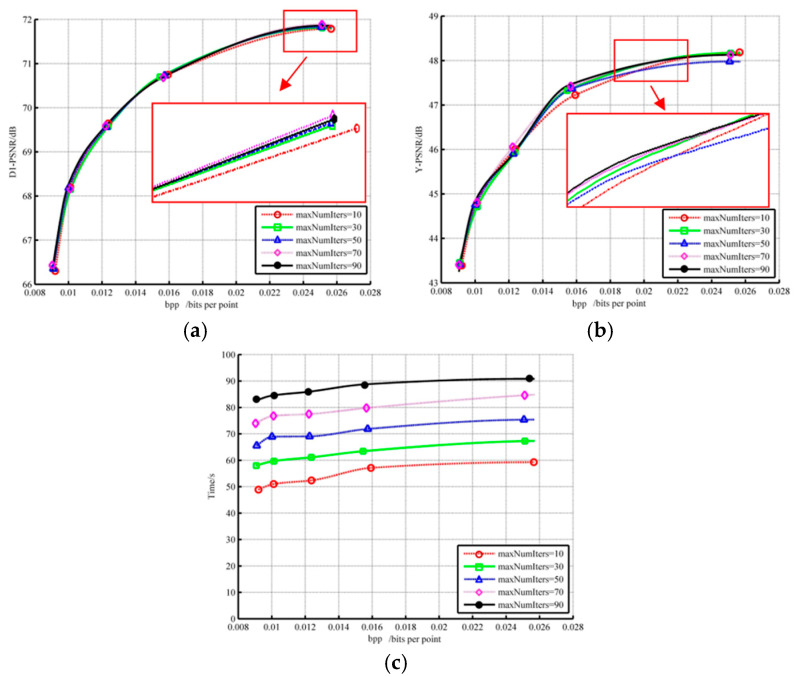
The influence of different numbers of iteration on (**a**) geometry D1 performance; (**b**) color component; (**c**) time.

**Figure 10 sensors-22-01262-f010:**
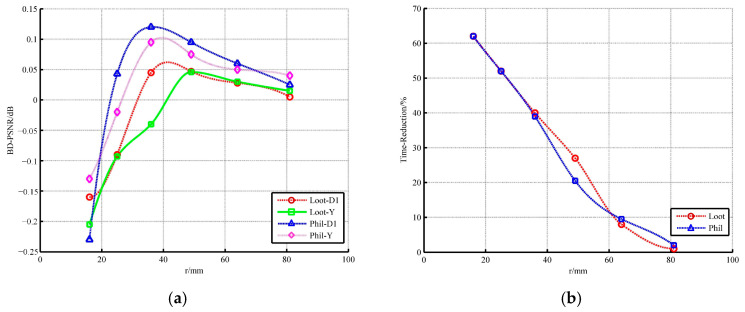
The influence of radius on (**a**) R.D. performance; (**b**) time.

**Figure 11 sensors-22-01262-f011:**
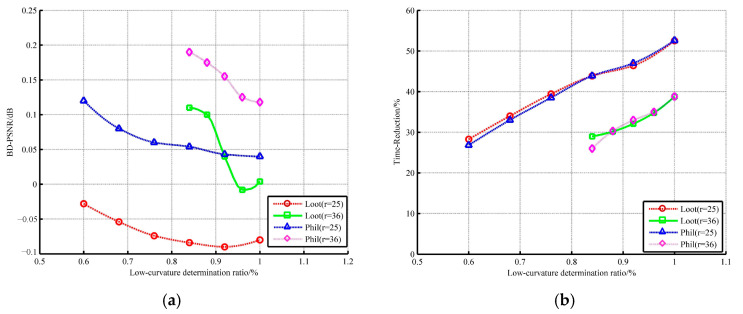
The influence of low-curvature determination ratio on (**a**) geometry performance; (**b**) time.

**Figure 12 sensors-22-01262-f012:**
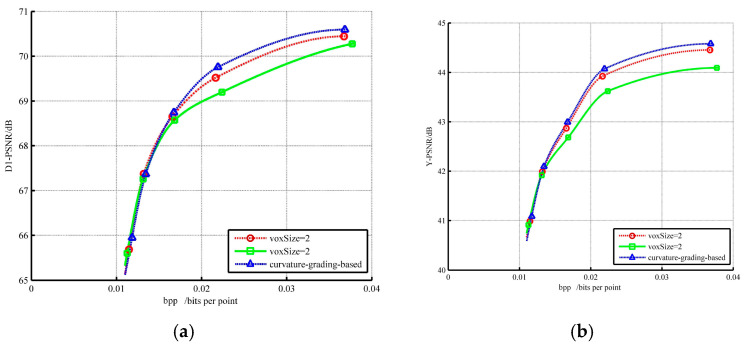
R.D. curves for *Redandblack*: (**a**) geometry performance; (**b**) color performance.

**Figure 13 sensors-22-01262-f013:**
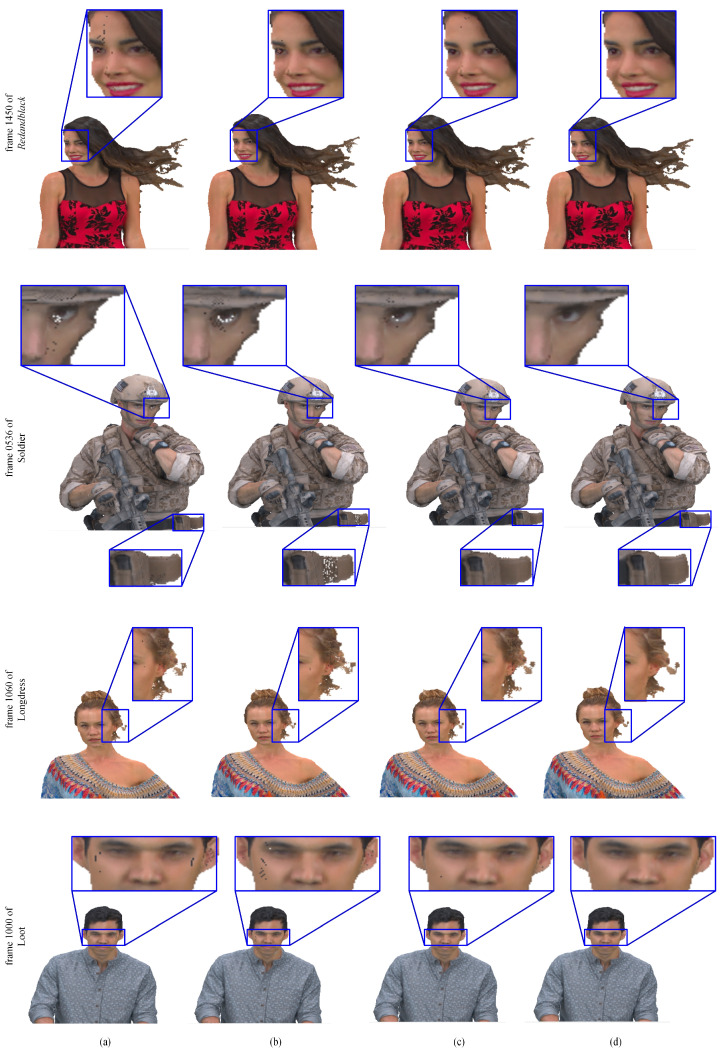
Results for subjective inspection in MPEG 8i dataset: (**a**) V-PCC with voxDim of 2; (**b**) V-PCC with voxDim of 4; (**c**) compressed by the curvature-grading-based refining method; (**d**) the original point clouds.

**Figure 14 sensors-22-01262-f014:**
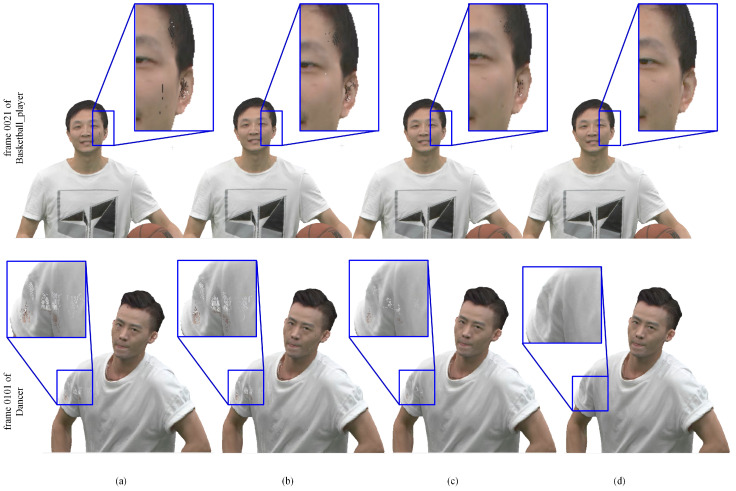
Results for subjective inspection in Owlii dataset: (**a**) V-PCC with voxDim of 2; (**b**) V-PCC with voxDim of 4; (**c**) compressed by the curvature-grading-based refining method; (**d**) the original point clouds.

**Figure 15 sensors-22-01262-f015:**
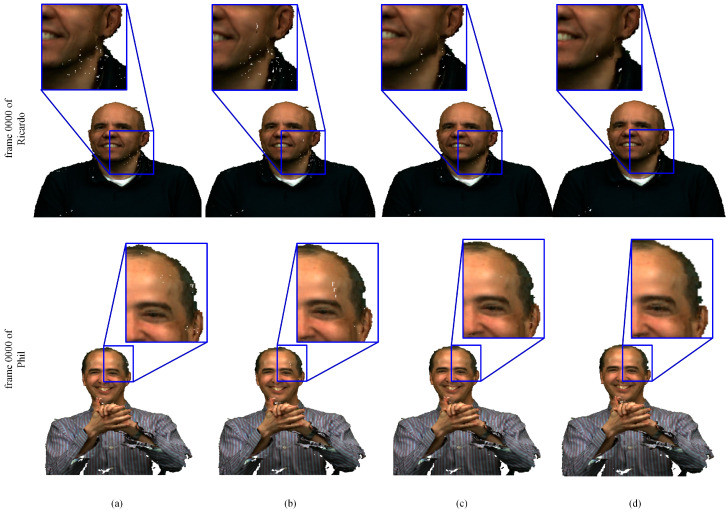
Results for subjective inspection in Microsoft Voxelized Upper Bodie’s dataset: (**a**) V-PCC with voxDim of 2; (**b**) V-PCC with voxDim of 4; (**c**) compressed by the curvature-grading-based refining method; (**d**) the original point clouds.

**Table 1 sensors-22-01262-t001:** The BD-rate, BD-PSNR, and runtime for V-PCC with voxel size of 4 in comparison with V-PCC with voxel size of 2.

Sequences	D1		Y		Time-Reduction (%)
	BD-Rate (%)	BD-PSNR (dB)	BD-Rate (%)	BD-PSNR (dB)	
Red	2.0689	−0.0868	1.3450	−0.0590	55.84
Soldier	4.9579	−0.1087	3.0096	−0.1402	52.34
Long	9.2006	−0.1443	2.0689	−0.0868	54.12
Loot	0.6707	−0.0183	11.7859	−1.9272	54.56
Ricardo	3.1759	−0.0592	2.2672	−0.2622	58.32
Phil	1.2486	−0.0488	6.0124	−0.1591	57.02
Basketball_player	0.8839	−0.0261	0.0016	−0.1750	50.71
dancer	0.0483	−1.2106	0.0651	−1.2304	46.90
Average	2.7819	−0.2129	3.3195	−0.5050	53.73

**Table 2 sensors-22-01262-t002:** The BD-rate, BD-PSNR in geometric component, and runtime for our method compared with V-PCC.

Sequences	voxelSize = 2			voxelSize = 4		
	BD-Rate (%)	BD-PSNR (dB)	Time-Reduction (%)	BD-Rate (%)	BD-PSNR (dB)	Time-Reduction (%)
Red	−1.7074	0.0269	34.50	−1.8712	0.1275	−47.18
Soldier	−1.0711	0.0421	31.93	−9.7654	0.4410	−57.97
Long	−0.3460	0.0250	33.62	−1.1740	0.0520	−54.12
Loot	−0.8159	0.0507	33.46	−5.5974	0.2262	−52.4
Ricardo	−1.4667	0.0559	36.62	−4.4987	0.1152	−49.35
Phil	−3.5791	0.1685	34.40	−6.7303	0.2155	−52.46
Basketball_player	−2.7974	0.2752	31.44	−2.3516	0.0471	−44.83
Dancer	−2.3710	0.0568	33.09	−2.9765	0.0616	−45.81
Average	−3.1455	0.0876	33.63	−4.3706	0.1608	−50.52

**Table 3 sensors-22-01262-t003:** The BD-rate and BD-PSNR in Y color component for our method in comparison with V-PCC.

Sequences	voxelSize = 2		voxelSize = 4	
	BD-Rate (%)	BD-PSNR (dB)	BD-Rate (%)	BD-PSNR (dB)
Red	−3.8787	0.0974	−4.9378	0.3130
Soldier	−4.5162	0.1814	−13.045	0.4375
Long	−0.0822	0.0354	−5.2017	0.2171
Loot	−0.3806	0.0632	−10.0006	0.4215
Ricardo	−6.7822	0.0407	−12.9192	0.2941
Phil	−4.4651	0.1443	−9.7481	0.3052
Basketball_player	−1.6300	0.2258	−1.5219	0.0577
Dancer	−2.4382	0.0878	−3.8152	0.1341
Average	−3.0217	0.1095	−7.6487	0.2725

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
