# Peer review of "A Method Based on Curvature and Hierarchical Strategy for Dynamic Point Cloud Compression in Augmented and Virtual Reality Systemâ€"

_sensors, 2022, doi:10.3390/s22031262_

Round 1
Reviewer 1 Report
This paper presents a video-based point cloud compression method in augmented and virtual reality system based on curvature and hierarchical strategy, and the experiments show that the proposed method works well in MPEG 8i dataset. However, there are still some discussions and modifications to be done before accepting.
1) 1. Introduction: “However, this method didn't use the correlation between adjacent frames.” As far as I know, some other works use the correlation between adjacent frames, such as 3D Motion Estimation and Compensation Method for Video-Based Point Cloud Compression.
2) 2. The Curvature-based hierarchical Dynamic Point Cloud Compression Method:you'd better to expound the contribution of the proposed method.
3) 3.1. Test Materials and Conditions: More complicated datasets should be performed to verify the validity, such as dataset with combination of figure and other objects.
4) 3.1. Test Materials and Conditions: “the proposed approach improves performance and efficiency obviously with an average time-consuming saving of 34.09%.” Does time-consuming take initial segmentation step and curvature-based hierarchical refinement step into account?
5) 3.4. RD Performance Evaluation: It would be better to enlarge Figure.12 that we can tell the difference.
Reviewer 2 Report
The article entitled « A Method Based on Curvature and Hierarchical Strategy for Dynamic Point Cloud Compression in Augmented and Virtual Reality System » proposes an improved point cloud compression method.
The proposed method is relevant and allows a clear improvement of the point cloud compression compared to existing approaches.
Overall the proposed method is interesting enough to substantiate its publication in the journal.
However, it may not be suitable for the "Intelligent Sensors" section as the article’s topic is focused on point cloud compression.
Regarding the structure of the article, the article generally follows the guidelines for article submission required by the journal.
The introduction presents the research context as well as the results and challenges of existing approaches for point cloud compression and introduces the proposed method. Section 2 presents the method in sufficient detail to be reproducible. Section 3 presents the experiments conducted to evaluate the effectiveness of the proposed method. Finally, section 4 summarizes the information contained in the article.
In view of the subject of the article, the title does not adequately describe its content, especially the part of virtual reality (or augmented reality) is not sufficiently present in the article to justify its presence in the title.
The summary effectively presents the content of the article.
Specific comment:
The introduction does not provide an overview of the experiments that were conducted nor does it provide an outline of the article.
Lines 26-27: Do you have any scientific references to support your following argument " the use of 3D point clouds to represent real-world scenarios in an immersible fashion has become increasingly popular in recent decades"
Line 36: You specify that it is important to compress the point cloud before using it for augmented reality or virtual reality applications; Why use a point cloud and not convert it to a mesh which is easier to simplify?
Line 88:The abbreviation AR/VR is used without prior explanation
Line 56: “Some scholars who are prominent in image compression call for the development of PCC standards”: This information does not sound very scientific but is rather a value judgment, I recommend you to delete them.
Lines 57-66: You present the approaches proposed by JEPG but you do not comment on the results they achieve or why they should be improved.
Line 88:The abbreviation R.D is used without prior explanation
Lines 92-97: The different steps presented are not justified. As it is, it is not clear why each of the steps is essential or whether they can be replaced by other methods.
Lines 102-115:
Le processus de segmentation n'est pas trivial à suivre. Une image l'illustrant permettrait de faciliter sa compréhension.
Line 114: Can you provide more information and clues on how the refinement is done?
The visualization of figures 1,3 and 4 can be improved by adding a visualization taking into account the normals in each point in order to have more contrast and thus allow us to see more details.
It is a pity that figure 12 is at the end of the article, it is relevant for the visualization of the results in section 3.
